# Nutrition and Healthy Aging: Prevention and Treatment of Gastrointestinal Diseases

**DOI:** 10.3390/nu13124337

**Published:** 2021-11-30

**Authors:** Neri Maria Cristina, d’Alba Lucia

**Affiliations:** 1Division of Gastroenterology, Geriatric Institute Pio Albergo Trivulzio, 20146 Milan, Italy; 2Department of Gastroenterology and Endoscopy, San Camillo Forlanini Hospital, 00149 Rome, Italy; luciadalba@tiscali.it

**Keywords:** healthy aging, malnutrition, gastrointestinal diseases

## Abstract

Nutritional well-being is a fundamental aspect for the health, autonomy and, therefore, the quality of life of all people, but especially of the elderly. It is estimated that at least half of non-institutionalized elderly people need nutritional intervention to improve their health and that 85% have one or more chronic diseases that could improve with correct nutrition. Although prevalence estimates are highly variable, depending on the population considered and the tool used for its assessment, malnutrition in the elderly has been reported up to 50%. Older patients are particularly at risk of malnutrition, due to multiple etiopathogenetic factors which can lead to a reduction or utilization in the intake of nutrients, a progressive loss of functional autonomy with dependence on food, and psychological problems related to economic or social isolation, e.g., linked to poverty or loneliness. Changes in the aging gut involve the mechanical disintegration of food, gastrointestinal motor function, food transit, intestinal wall function, and chemical digestion of food. These alterations progressively lead to the reduced ability to supply the body with adequate levels of nutrients, with the consequent development of malnutrition. Furthermore, studies have shown that the quality of life is impaired both in gastrointestinal diseases, but especially in malnutrition. A better understanding of the pathophysiology of malnutrition in elderly people is necessary to promote the knowledge of age-related changes in appetite, food intake, homeostasis, and body composition in order to better develop effective prevention and intervention strategies to achieve healthy aging.

## 1. Introduction

Healthy aging has been identified by the World Health Organization as a work priority, and a policy framework has been designed to improve older people’s well-being and ensure they are able to live independently in society [1].

Aging is characterized by a reduced regenerative and adaptive capacity resulting in an easier development of morbidity; one’s prerequisite for wellbeing and having a good quality of life is to maintain an adequate nutritional status [2,3].

Nutritional well-being is a fundamental aspect for the health, autonomy and, therefore, the quality of life of all people, but especially of the elderly.

It is estimated that at least half of non-institutionalized elderly people need nutritional intervention to improve their health and that 85% have one or more chronic diseases that could improve with correct nutrition [4,5].

The rapid globalization of society has resulted in multiple forms of malnutrition, especially in the most fragile populations such as old people [6,7].

This concept, termed the “double burden of malnutrition” is referred to by the WHO as “characterized by the coexistence of undernutrition, along with overweight, obesity or diet-related noncommunicable diseases, within individuals, households and populations, and across the life-course” [8].

Although prevalence estimates are highly variable, depending on the population considered and the tool used for its assessment, malnutrition in the elderly has been reported from 3% in the community setting up to 50% [3,9]; in particular, its incidence ranges from 12% to 50% among the hospitalized old patients and from 23% to 60% among the institutionalized older adults [10,11,12,13].

Weight loss, due to a lack of macronutrients or increased catabolism, represents the initial step of a catabolic process, which results in a high morbidity and mortality rate.

Older patients are particularly at risk of malnutrition due to multiple etiopathogenetic factors (changes in metabolism and organic function, a decline in physiological reserves associated with aging processes, chronic diseases, and drug polypharmacy), which can lead to a reduction or utilization in the intake of nutrients, a progressive loss of functional autonomy with dependence on food, and psychological problems related to economic or social isolation, e.g., linked to poverty or loneliness [14].

Although gastrointestinal function undergoes modest changes with aging [15], it is possible that there may still be alterations in the intake and assimilation of nutrients; it follows that a geriatric patient is defined on the basis of their fragility and the presence of chronic diseases, rather than by a specific age.

Gastrointestinal changes in aging involve motor function and therefore intestinal transit, mechanical breakdown of food and his chemical digestion. These alterations progressively lead to the reduced ability to supply the body with adequate quantities of nutrients, with the consequent development of malnutrition [16,17,18].

A decreased appetite among healthy older people may be due to a decreased stomach capacity and to a reduced gastric emptying due to its impaired relaxation, perhaps due to the fluctuations in the production and secretion of various enterohormones, such as higher levels of cholecystokinin and a lower concentration of ghrelin, which can determine to early satiety after a meal. Furthermore, aging can lead to a reduction in the number of taste bods, resulting in deterioration of the sense of smell.

Finally, these conditions can lead to a loss of appetite with a reduction in meals consumed regularly and consequent malnutrition [19], which, in turn, increase the risk of developing a number of diseases, in particular affecting the nervous, musculoskeletal, cardiovascular, immune, and skin system [20,21].

A better understanding of the pathophysiology of malnutrition in elderly people is necessary to promote the knowledge of age-related changes in appetite, food intake, homeostasis, and body composition in order to better develop effective prevention and intervention strategies to achieve healthy aging [22,23,24,25].

## 2. Definition and Causes of Malnutrition

Malnutrition is described by different organizations as the condition of being poorly nourished, caused by a lack of one or more nutrients (undernutrition) or an excess of nutrients (overnutrition) [7,26,27].

Overnutrition is due to the excessive intake of nutrients, with the accumulation of body fat defined as overweight or obesity, with a prevalence from 19 to 21% of women and men and from 10 to 13% in rural areas, that compromises health, leading to chronic disabling diseases, especially cardiometabolic ones [27]. An underestimated phenomenon is sarcopenic obesity [28], characterized by a reduced lean body mass and increased fat mass. In light of this concept, it is very important to evaluate muscle mass regardless of weight loss.

In this review, we do not deal with overnutrition, but focus on the other, more taken into consideration aspect of malnutrition: undernutrition.

A definition of malnutrition universally accepted has not yet defined [29]. According to The European Society for Clinical Nutrition and Metabolism (ESPEN-2015), malnutrition can be defined as “a state resulting from lack of intake or uptake of nutrition leading to diminished physical and mental function and impaired clinical outcome from disease” [30].

In 2017, for geriatric patients, the guidelines on the Enteral Nutrition in Geriatrics by the European Society of Clinical Nutrition and Metabolism (ESPEN) established that malnutrition is characterized by the presence of either weight loss (which reflects a catabolic state) and/or a low BMI, representing depleted physiological stores [31].

In 2019, the Global Leadership Initiative on Malnutrition (GLIM) [32,33] was published, the result of a consensus of more than 70 national scientific societies represented by the four leading clinical nutrition societies (ESPEN, ASPEN, FELANPE, and PENSA), in order to develop criteria for malnutrition that could be used in all clinical settings on a global scale. It considers three phenotypic criteria—(1) weight loss, (2) a low body mass index, and (3) a reduced muscle mass—and two etiological criteria—(1) a reduced food intake or assimilation and (2) inflammation (disease/acute injury or related to chronic diseases). The diagnosis of malnutrition is based on the presence of at least one phenotypic and one etiological criterion, with the subsequent use of different thresholds of the criteria for the classification of its severity of malnutrition [32,33].

Furthermore, there are numerous ongoing evaluation studies on the GLIM criteria.

We know three types of weight loss, that are wasting, cachexia, and sarcopenia. 

Wasting is due to an insufficient dietary intake and leads to involuntary weight loss.

Cachexia is due to induced catabolic processes, with the production of proinflammatory cytokines such as interleukin-1 (IL-1), tumor necrosis factor alpha (TNF-alpha), IL-6, and others playing a predominant role; its main consequence is a reduction of lean mass and body cell mass [34]

Sarcopenia was defined by the European Working Group on Sarcopenia in Older Persons (EWGSOP) as a loss of muscle mass in combination with a loss of muscle strength or physical performance; these two entities often occur together, this has led to the new term “malnutrition sarcopenia syndrome” [35].

The etiology is poorly understood, but an important role is due to physical inactivity, the induction of a proinflammatory response, and the dysregulation of anabolic hormones, such as testosterone or growth hormone [7,27,30].

Depending on the type of malnutrition, protein catabolism can be pronounced. Malnutrition leads to protein catabolism with a rapid wasting of skeletal muscles, a lower muscle mass, and reduced muscle strength. At the same time, there could be a reduced dietary protein intake [36,37] with a decreased bone mineral mass [17,18]. The result is an impaired musculoskeletal function, increased disability, and with reduced physical performance an increased risk of falling, with a greater risk of osteoporosis and osteoporotic fractures [38].

Malnutrition also leads to an impaired immune function, with reduced cell-mediated immunity, and an increased risk of infection and delayed healing [39].

The mechanisms that lead to malnutrition in elderly people are complex, butcan ultimately result from starvation, disease, or advanced ageing, alone or in combination. 

The main cause is to be found in a reduced and/or unbalanced dietary intake, due to many causes that can be divided in three main categories: physiological, psychological, and social. In developed countries, the main factor of malnutrition is disease [40].

The examples for each category are summarized in the Table 1.

## 3. Pathophysiology of Undernutrition

Age is able to induce changes in the function of all organs and human physiological processes [30]:➢Intestinal function with a reduction in sensory perceptions, salivation, chewing, absorption of nutrients, and lactose tolerance;➢Brain function with cognitive impairment and Alzheimer’s disease;➢Body composition with loss of lean mass (especially in skeletal muscle tissue, a phenomenon known as sarcopenia) and increase in fat mass with variation in its distribution;➢Balance of fluids;➢Bones and joints with osteoporosis and/or arthritis with a consequent increased risk of falls and fractures;➢Metabolism with type 2 diabetes mellitus and dyslipidemia;➢The cardiovascular system with various diseases;➢Cell growth with cancer (cancer in the elderly exhibits a slower growth because they show a slower rate of cell development than in young people) [41].

## 4. The Aging Gastrointestinal Tract

The gastrointestinal tract represents the primary and largest area of contact with environmental factors and ingested pathogens.

The regular functioning of the gastrointestinal tract is essential for maintaining good health of the body, as it guarantees the absorption of nutrients and drugs, and protection against external pathogens.

Futhermore, the multi-organ system that composes the gastrointestinal tract has large reserve capacity, and thus there is little changes in gastrointestinal function because of aging in absence of specific disease [15].

Nevertheless, some older older people can show a reduction of gastrointestinal functions, including immune function [42,43], with a greater frequency of bacterial and viral gastrointestinal infections compared to young people. Aging is associated with structural and functional alterations of mucosal defense, an increased oxidative stress, reduced ability to generate protective immunity, and increased incidence of inflammation and autoimmunity [44].

Particularly, the gatrointestinal mucosal layer represents the first line of defense against external pathogens and is equipped with multiple defense mechanisms, including the secretion of alkaline mucus, bicarbonate and antimicrobial peptides, epithelial intercellular tight junctions, antioxidants, autophagy, and antimicrobial peptides, and the innate mucosal immune system [44].

Mucus secretion performs multiple functions, such as protecting epithelial cells from pathogens, acids, digestive enzymes, and abrasion from food residues [45].

Aged mucosal surfaces are susceptible to lesions which appear to be receptive to differential levels of sex hormones and so it could contribute to the sexual dimorphism observed in response to lesions on the mucosal surface, before and after menopause [46].

Furthermore, in the elderly, the response to infection is often exacerbated by malnutrition [47]; in fact, low-grade chronic inflammation can derive from the intestinal environment but with consequences even at a distance from the intestine, leading to fragility, sarcopenia, and degenerative disorders affecting various organic systems, including the central nervous system.

## 5. Gastrointestinal Functions

### 5.1. Chewing Activity and Esophagus

The chewing function is due to two main factors: the masticatory forces linked to the number of antagonistic teeth and the quantity/quality of saliva (linked to salivary enzymes such as alpha-amylase or lipase).

A major problem in the elderly can be poor oral health and dental conditions, which can lead to difficulty chewing and dry mouth with a reduced food intake [48,49,50,51].

With age, there is a decrease in taste linked to an increase in the thresholds for the basic tasters—bitter, salty, sweet, acidic—which can lead to a loss of appetite and help reduce the energy intake, favoring the development of malnutrition [51,52,53,54].

Disorders of the lower esophageal sphincter function are also common in asymptomatic elderly people, even with an overall esophageal function being well preserved, and may contribute to the development of dysphagia.

In older dysphagic patients, the basal lower esophageal sphincter is increased and swallowing-induced relaxation is incomplete, as demonstrated by high-resolution manometry studies [55].

### 5.2. Stomach

Gastric emptying plays an important role in the kinetics of nutrient absorption, which in turn is mainly controlled by the feedback of the neural and humoral signals which are generated by the interaction of the nutrients themselves with the small intestine [56].

Liquid and solid meals display different gastric emptying rates after ingestion, the first being faster and the second slower.

Healthy aging is characterized by a modest slowing of gastric emptying of both solids and liquids, especially in the initial phase, and with protein nutrients, but emptying generally remains within the normal range [57].

The slowing of gastric emptying has implications for appetite regulation and could potentially contribute to the anorexia of aging.

In addition, in old age, blood pressure may decrease in response to meal (postprandial hypotension); in fact, post-prandial blood pressure may depend on the regulation of the splanchnic blood flow, gastric distension, and the release of gastrointestinal hormones, consequent to both gastric emptying rate and the enteric response to ingested nutrients [58].

This represents a relevant clinical problem, especially if associated with a series of symptoms that negatively affect the quality of life and can increase mortality, such as syncope, falls, angina, dizziness, nausea, dizziness, and/or visual disturbances; patients with type 2 diabetes and Parkinson’s disease are particularly at risk [15,59,60].

Gastric acid secretion is reduced with age, mainly linked to atrophic gastritis, but also with a prolonged use of drugs such as pump inhibitors; it represents an important non-immunological defense against external pathogens and its reduction (hypochloridria) can predispose the small intestine to bacterial overgrowth, causing a low body weight and symptoms such as diarrhea, a reduced absorption of nutrients, including folate, vitamins B 12, B 2, B 6, and fiber [15,61,62].

Atrophic gastritis type A of the body is associated with an autoimmune disease, pernicious anemia, characterized by the presence of antibodies for parietal cells and intrinsic factors, present in <5% of patients over >60 years; the consequence could be a deficit of B1 absorptions, being rare since the stomach has a large reserve of intrinsic factor.

Atrophic gastritis type B of the antrum and gastric body is due to environmental conditions (the most frequent cause is Helicobacter pylori infection, a risk factor for the incidence of gastric cancer) and is present in >25% of people aged >60 years.

Furthermore, there may be a reduced absorption of iron and calcium, linked to gastric pH [21,62,63].

With age, the risk of gastropathy is increased due to toxic agents such as non-steroidal anti-inflammatory drugs (or ethanol, aspirin, etc.), also due to a reduction in protective factors such as prostaglandins and the secretion of bicarbonate, sodium, and liquids [56,64,65].

### 5.3. Small Bowel

Small bowel transit time in young adults ranges from 2 to 6 h.

The patterns of motility and the transit rate appear to be maintained in the small intestine during aging, although the propagation velocity of phase three of the migrating motor complex is slower in the elderly.

The intestinal epithelium is characterized by a rapid turnover (every 2 to 6 days); tissue regeneration is compromised in the aged population, with consequent mucosal atrophy and damage to epithelial protein; however, the total surface area available for absorption in the small intestine is, therefore, not so affected by aging and absorption, in general, is not deeply compromised [47,66].

The transit time of the small intestine in young adults varies from 2 to 6 hours; during aging, there does not seem to be any variations, both in terms of intestinal motility and transit, although the propagation velocity of phase three of the migrating motor complex is slower.

Intestinal permeability, assessed through the lactulose/mannitol test, also remain unchanged in old age [67], as well as the carrier function, which is not affected by aging in a relevant way [40].

### 5.4. Colon

The high prevalence of motility disorders in the elderly, especially constipation and fecal incontinence, could be due to neuronal degeneration in the enteric nervous system of the colon [68].

In reality, clinical studies are not so specific: in fact, Metcalf et al. [69] found a regular transit time in different segments of the colon of healthy elderly, while Madsen and Graff [70] reported a significant increase in colon transit time in the same population. Additionally, environmental factors, such as physical inactivity, can significantly affect colon transit time.

Therefore, it is not possible to establish a specific role of aging on the intestinal transit time.

### 5.5. Pancreas

Pancreatic exocrine secretion decreases in old age, but this decline may not be enough to cause maldigestion in advanced age [71].

Nevertheless, some studied showed that in pancreatic secretions recorded from the duodenal collection of subjects older than 65–70 years, the presence of bicarbonate and enzymes (lipase, chymotrypsin, and amylase) was significantly reduced compared to young controls, both in regards to the secreted volume and the concentration of enzymes [72].

Furthermore, an ample population study (50–75 years) reported in old age an increase of fecal elastase-1, that is a marker of exocrine pancreatic dysfunction, thus attesting an exocrine pancreatic insufficiency [73].

Finally, to support these data, an increase in pancreatic atrophy, lobulation, and fatty degeneration was recorded using magnetic resonance imaging during aging [74,75].

However, these data supporting the pancreatic exocrine secretion decline may not be enough to cause maldigestion in advanced age [60].

Regarding the pancreatic endocrine function, the number and mass of B cells are relatively well preserved in the endocrine pancreas of non-diabetic old people [76].

However, aging is associated with insulin resistance, resulting in increased fasting and postprandial blood glucose [77].

### 5.6. Microbial Digestion

With aging, physiological changes in the gastrointestinal tract are associated with major changes in the gut microbiota, with a reduced microbial stability and diversity and rearrangements within the Firmicutes and Bacteroidetes phyla.

In particular, low health-promoting bacteria, especially short-chain fatty acid-producing bacteria (SCFA) such as bifidobacteria, associated with an increased overgrowth of pathobionts (e.g., streptococci, staphylococci, enterobacteria, and enterococci anaerobes), can result in an overall decrease in saccharolytic production. At the same time, it seems to increase the proteolytic potential, resulting in a predominant putrefactive metabolism, with a gradual decline in immune system function and, therefore, contributing to an augmented risk of infection and frailty [78,79,80,81].

This altered composition of microbiota significantly correlates with a proinflammatory status, and with a higher incidence of disease, frailty, co-morbidity, and undernutrition [44].

The reduction of short-chain fatty acids has important repercussions on the organism. They are derived from the undigested oligosaccharides and monosaccharides, which undergo bacterial degradation and fermentation, so as to be transformed into short-chain fatty acids (SCFA), i.e., acetate, propionate, and butyrate. SCFA are then almost completely absorbed in the colon, and used as energy for the colonocytes or transported to various peripheral tissues, influencing the human metabolism and exerting important anti-inflammatory and antineoplastic effects, promoting the formation and protection of the intestinal barrier from the harmful action of lipopolysaccharide (LPS); furthermore, they represent the most important drivers for microbiota change in the elderly [82,83].

Centenarians show a different and unique microbiota composition as compared to young adults and the elderly; in fact, both seem to have a similar bacterial structure, with a high composition of Bacteroidetes and Firmicutes and a minor population of Actinobacteria and Proteobacteria [84].

Furthermore, there is a shift of the Firmicutes population to a low diversity in terms of species composition, with an increase in Bacilli and a decrease and rearrangement of some cluster of Clostridium (i.e., the butyrate-producing bacteria *Ruminococcus*, *Roseburia Eubacterium Faecalibacterium prausnitzii*) [84]. This is very important because butyrate is a major energy source for the enterocytes and has a significant anti-inflammatory role.

Finally, aged people with high levels of the mucin degrading Akkermansia muciniphila were found in comparison with the young adults [84].

Furthermore, in semi-supercentenarians, i.e., those aged 105–109, in comparison to adults, the elderly, and centenarians, a decrease was found in symbiotic bacteria, belonging mainly to the dominant *Ruminococcaceae*, *Lachnospiraceae,* and Bacteroidaceae families, as well as an increase in opportunistic bacteria along with age [85].

Ultimately, it is very important to emphasize and promote a correct diet and good life-style in light of the fact that they are considered to be the most important drivers for microbiota change in the elderly [86].

### 5.7. Intestinal Barrier and Immune System

In the elderly, modifications of the mucus could explain the reduced ability of bifidobacteria to bind to the mucosa [87], but, apart from *H. pylori*-positive subjects, the thickness of the mucus layer is not altered [88].

Aging is associated with “immunosenescence”, which is also a progressive reduction in the mucosal immune response of the intestine, linked to a reduced ability of the aging immune system to provide a tolerance towards antigens and a reduced production of immunoglobulin-specific antigen A, which represents the most important defensive mechanism of the mucosal immune response [89].

Finally, the lack of oral tolerance in old age can be explained by a decrease in the number and functionality of dendritic cells, which have the task of presenting antigens to immunocompetent B and T lymphocytes [90]; however, in Peyer’s patches, the density of mononuclear phagocytes did not vary with aging.

Table 2 shows the functional and/or organic changes that gastrointestinal tract may undergo with aging and the possible nutritional repercussions.

## 6. Adverse Consequences of Undernutrition

Aging associated with malnutrition leads to a progressive deterioration of health in the elderly, with a consequent decrease in physical and functional abilities and a greater vulnerability of the patient, dependence in the activities of daily life, a poor quality of life, and greater morbidity and mortality [91].

Malnutrition also negatively affects the outcome of therapies because it reduces the immune response with a consequent increase in infections and delay in healing, reduces muscle mass, favoring inactivity and dependence, with psychological consequences leading to depression with a relative possible loss of appetite [92].

Elderly subjects tested with BMI, weight loss and food intake have consistently showed an association between mortality and malnutrition or nutritional risk [93,94].

### 6.1. Nervous System

Neurodegenerative diseases and altered cognitive functions may be the cause, but also the result, of malnutrition.

In particular, an association between cognitive deficits and/or depressive symptoms and low levels of vitamin B6, folate, vitamin B12, and polyunsaturated fatty acids (PUFAs) was found in old people [95].

The brain has an high metabolic activity; therefore, it is often subject to oxidative stress with possible damage to neural tissue. A prevailing theory is that at the base of neurodegenerative disorders, such as Alzheimer’s disease and Parkinson’s disease, there may be a biological mechanism linked to oxidative damage with consequent neuronal inflammation [96].

Futhermore, antioxidant nutrients may, therefore, play more relevant roles in the aging brain than in other organs because of the reduction in the number of antioxidant enzymes required for neuronal protection.

This is why a reduced intake of antioxidant nutrients, typical of malnutrition or poor-quality diets, can, therefore, negatively affect cognitive function [97].

The only micronutrient with antioxidant properties promising to be protective on cognitive activities is vitamin E. Indeed, prospective epidemiological studies on dietary vitamin E supplementation have consistently shown statistically significant inverse associations with dementia, Alzheimer’s disease, and cognitive decline [97,98]. Vitamins B9 and B12 are also nutrient cofactors able to modulate neurocognitive development and neurodegeneration [99,100]. Blood levels of vitamin B12 are often low even among the elderly with an adequate intake [40].

### 6.2. Musculoskeletal System

In healthy adults, muscle accounts for over half of the total organic protein.

In the elderly, the muscles undergo a reduction in the number, size, and contractility of muscle fibers, as well as a fat infiltration of the skeletal muscle.

These deteriorations, associated with increased body fat, systemic low-level inflammation, and oxidative stress, contribute to further changes in the musculoskeletal system that could lead to sarcopenia, osteoporosis, weight loss, malnutrition and frailty [101].

Frailty represents a geriatric syndrome characterized by a lesser homeostatic capacity and a reduction in physiological functional reserves, which lead to a particular vulnerability and consequent adverse health outcomes, including falls, fractures, and an increased mortality.

Sarcopenia was defined by the European Working Group on Sarcopenia in Older Persons (EWGSOP) as a loss of muscle mass in combination with a loss of muscle strength or physical performance [102].

Sarcopenia, osteoporosis, and frailty are worsened by malnutrition and specific nutrient deficiencies, in particular, protein, antioxidant vitamins, minerals, and fatty acids [103].

Vitamin D represents a key function in bone and mineral metabolism, and its deficiency leads to metabolic bone disease through a reduced intestinal calcium absorption with secondary hyperparathyroidism, consequent impaired mineralization with an increased bone reabsorption [104].

According to epidemiologic studies, protein intake correlates positively with body mineral density and negatively with the rate of bone loss.

To retrieve and maintain muscle function with a lean body mass, the PROT-AGE Study Group recommends an average daily intake of at least 1.0–1.2 g protein per kilogram of body weight. Furthermore, it is necessary to integrate increasing dietary protein into an energy-controlled dietary plan for weight management [105,106].

Calcium is an important architectural component of bones and, thus, very important for bone health. An inadequate calcium absorption increases the concentration of the parathyroid hormone and so can lead to an increased bone reabsorption [107].

### 6.3. Cardiovascular Disease

In general, heart disease and atherosclerosis are correlated with overnutrition rather than undernutrition.

The heart is a muscle that becomes vulnerable to numerous micronutrient deficiencies such as vitamin A, vitamin C, vitamin E, thiamine, B vitamins, vitamin D, selenium, zinc, and copper, resulting in heart failure (even up to myocardial infarction); a serious consequence is characterized by right heart failure with an intestinal edema and related malabsorption, which can lead to cardiac cachexia with severe malabsorption syndrome and important weight loss [108,109].

In addition, cardiac patients tend to follow diets that sometimes have highly restrictive indications. Full adherence to such diets induces malnutrition and major nutritional deficiencies, which, ultimately, aggravate heart failure and lead to physical deterioration [110].

Additional mechanisms to aggravate the nutritional status of patients with heart failure include reduced intestinal absorption linked to intestinal edema, increased urinary protein loss due to drug therapy, and greatest oxidative stress.

In conclusion, the possible nutritional deficiencies of micronutrients associated with cardiovascular disease can significantly increase morbidity and mortality.

### 6.4. Immune System

Immunosenescence is represented by a reduction in the ability of the aging organism to counteract the attacks of external pathogenic agents through an efficient immune response and is expressed through a dysfunction of T cells with aging, due to a reduced proliferation of T lymphocytes and an impaired T-helper activity, with consequent altered cell-mediated and humoral responses.

Hence, malnutrition with a reduced nutrient intake has repercussions on the immune system, resulting in a reduction in the total lymphocyte count, T cell proliferation, and interleukins.

Furthermore, old age is also represented by an increase in low-grade systemic inflammation (“inflammaging”), with a greater development of chronic inflammatory diseases [111,112].

The aging of the immune system is characterized by a decline in acquired immunity, as opposed to innate immunity which remains unchanged, if not enhanced [113].

The nutritional status is, therefore, an important factor influencing the T cell response, in addition to the pathogenic load, and malnutrition contributes significantly to the immunological defects found with age.

Dietary lipids are precursors of eicosanoids, prostaglandins, and leukotrienes; therefore, they have a significant influence on the immune system [114].

In addition to lipids, the main nutrients that influence immune function and, therefore, the host response to infections, are water-soluble vitamins (B6, folate, B12, and C) and, among the fat-soluble ones, vitamins A, D, and E [115].

In particular, vitamin D plays an important role, especially in the mechanisms mediated by Toll-like receptors; therefore, the evaluation of vitamin D is particularly relevant in frail elderly people who have limited sun exposure and who do not receive supplements.

Other trace elements present in the diet that are involved in immune function are selenium, zinc, copper, and iron [116,117].

Malnutrition in the elderly is also responsible for a reduced response to vaccines [118,119].

### 6.5. Skin System

Malnutrition increases the risk of the onset of pressure sores and infections, which also delay wound healing as they prolong the inflammatory phase, and alter the synthesis and proliferation of fibroblasts and collagen [120,121,122]. Pressure sores are an important and frequent cause of morbidity and mortality in old age.

## 7. Dietary Solutions for the Aging Gastrointestinal Tract

When a risk of malnutrition is diagnosed, it is essential to use measures to identify and correct risk factors and, subsequently, introduce nutritional support [9,123,124].

Dietary guidelines against malnutrition in the elderly.

The European Food Safety Authority (EFSA) established Dietary Reference Values (DRV) for the total carbohydrates and dietary fiber, fats, water, and protein for the European population [125,126,127].

In 2013, the EFSA reviewed nutritional recommendations, considering age categories of 60–69 years and 70–79 years. [128].

A daily intake of 30–40 kcal/kg and 1.0–1.5 g protein/kg body weight was proposed, in order to reduce the risk of protein–energy wasting and frailty in Europe, whereas the exact recommended values depend on the health status [124,129].

Muscle protein synthesis can be promoted by protein pulse feeding and the consumption of fast proteins (such as whey protein) to prevent sarcopenia in older adults [130].

Data from literature recommend that old people should consume a balanced diet and that micronutrient supplements should only be given when the food intake is too low, although several reviews point out that the elderly population is at an increased risk for vitamin and mineral deficiencies, especially vitamins A, B1, B2, B12, D, E, K, calcium, and potassium, although their energy intake is within recommendations [123,131].

The reasons for these deficiencies may be due not only to a reduced amount of food, but also to the poor choice and lack of variety of food, to an impaired intestinal absorbent function, to drugs that can interfere with the metabolism of micronutrients, to an inflammatory state that can reduce their plasma reserves.

Micronutrients have a multifactorial role in various metabolic processes (cell proliferation and growth, immune function, and genomic stability) and their deficiency is involved in many age-related pathologies [72,132].

However, there are important oral nutritional support strategies needed to address established malnutrition or for individuals at risk of malnutrition [123,133].

The first step is, therefore, represented by increasing the intake of protein and energy from food in the diet.

Oral food supplements (ONS) (sip feeds) can be used as adjuncts to nutritional management.

In this sense, the main measures include:To increase the number of meals to at least three a day, interspersed with snacks;To use fortified foods such as enriched bread, yogurt, or pasta;To increase nutrient density by adding traditional foods with milk powder or concentrate, grated cheese, eggs, fresh cream, and nuts;To hire nourishing liquids such as milk drinks, juices, and smoothies;To introduce ONS in the case of specific diseases associated with malnutrition, assumed as snacks or added to meals;To prepare foods with textures suitable for the oral health of elderly patients [124,134,135].

Food modifications: include adjustments aimed to make foods safer, more palatable, and energetic; they consist of changes in the content of micro and/or macronutrients, in their texture or flavor, in the organoleptic enhancement, or in the avoidance of specific allergens.

Additional nutrients can be added to food to increase its energy content and/or nutrient density (fortified or enriched food) or to yield specific health benefits (functional food).

Oral nutrition supplements (ONSs) (sip feeds): they provide both macro- and micronutrients which can be administered as liquids, or as semi-solids, or powders that can be prepared as drinks, or added to drinks or foods.

Based on expert consensus data, these supplements should provide at least 400 kcal and a minimum of 30 g of protein per day, and should be administered to all older people with a risk of malnutrition, in addition to dietary support; it is necessary to evaluate the benefits and possible compliance with a monthly assessment, to adapt the best supplements, and thereby tailoring the ONS-type to the elderly patient’s characteristics [30,136].

Tools for nutrition assessment.

Detection nutritional screening is very important and essential for the assessment of malnutrition risks of aged people and to establishing a correct dietary program as soon as possible [137].

Among the several screening tools that have been suggested for monitoring the nutrition assessment in the elderly, the most widely used and studied is the Mini Nutritional Assessment (MNA) in its long or short form.

With an appropriate adjustment of the cut-off point for different populations, the MNA and MNA-SF have been used and validated worldwide.

In 2002, ESPEN developed the ESPEN Guidelines for Nutrition Screening 2002 to provide guidelines on nutritional risk screening through tools applicable to different situations and based on evaluated evidence; these guidelines are still used [138].

Malnutrition Universal Tool (MUST) for adult community residents.Nutritional Risk Screening (NRS) for elderly hospitalized.Mini Nutritional Assessment Short Form (MNA) in the elderly.

The determination of the nutritional status in the elderly is performed through the mini nutritional assessment (MNA), the most frequently instrument used for this purpose.

It includes questions associated with physical and mental features and a questionnaire on nutrition.

MNA has a high predictive value characterized by its association with negative outcomes in terms of health and mortality.

MNA is divided into 18 questions grouped into four categories: anthropometry, general status, dietary habits, and nutrition states with self-perceived health. The summation of the points (maximum 30) allows an assessment of the nutritional status (score > 24 points: good status; 17 < score < 24 points: risk of malnutrition; score < 17 points: malnutrition).

MNA is considered the gold standard of nutritional assessments since it is adapted to the elderly, relatively easy to use, and highly sensitive [139]. However, it is associated with a high risk of “over-diagnosing” malnutrition in old age, and some authors have suggested integrating MNA with GLIM criteria to overcome the problem [33].

A multidisciplinary team, such as geriatricians, gastroenterologists, dietitians, and other professionals, when necessary, should be assembled to treat malnutrition in older patients.

The nutritional intervention, lasting at least three months, should work on the treatment of the underlying cause of malnutrition and on the improvement of the nutritional status, monitoring weight changes and the estimation of food intake, which are key aspects to assess the effects of the intervention [124,140].

Impact of major food products on malnutrition.

### Dairy Products

In old people, dairy products represent an important source of energy, protein, vitamins (A, B2, B5, B9, B12), minerals (calcium, magnesium, phosphorus, and zinc), and cholesterol; furthermore, some important nutrients, such as vitamin B9 and vitamin D, can be administered by fortifying milk [141,142,143].

Furthermore, probiotic yogurt showed a reduction in mutagenicity in the intestinal tract after consumption, as well as anti-inflammatory activity.

The immune system is positively influenced by dairy products containing bacteria, such as fermented milk or yogurt with added probiotic strains and/or lactic acid bacteria; data from literature show positive effects of these products in a series of pathologies such as Clostridium difficile infection, infectious airway diseases, and the common cold [144,145], as well as strengthening vaccination protocols [146,147], and reducing the development of osteoporosis when integrated with calcium and vitamin D [148].

This concept has been reiterated by the European Society for the Clinical and Economic Aspects of Osteoporosis and Osteoarthritis (ESCEO), which recommend in frail elderly patients at a high risk of falls and fractures the daily consumption of food products fortified with calcium and vitamin D (such as yogurt or milk) [147].

Dairy products, in particular milk proteins, are also rich in branched-chain amino acids (BCAAs), in particular leucine, which are necessary for muscle protein synthesis; thus, preventing the onset of sarcopenia in the elderly [105,149].

Other clinical indications are constipation, relieved by yogurt supplemented with dietary fiber, non-viral gastroenteritis, where probiotic fermented milk can alleviate symptoms [150,151,152], and dental health that can be improved by milk intake, supplemented with fluoride and probiotics [153].

Finally, dairy products provide an important matrix for introducing the flavors preferred by the elderly in order to deliver nutrients tailored to their nutritional needs; in fact, the taste of food is very important as it can induce changes in preferences for food consumption and so influence food intake [154].

Meat products

Older people tend to significantly reduce meat consumption for various reasons such as a difficulty in chewing, loss of smell and/or taste, and loss of appetite.

Unfortunately, old people are characterized by an increased protein requirement despite a reduction in their physical activity [155].

Meat has remarkable nutritional properties, including indispensable and essential amino acids (IAA), high levels of B vitamins and minerals, and is also useful for preventing possible deficiencies of vitamins and minerals (vitamin B12, iron, zinc, and selenium) when combined with other foods that are part of a healthy diet.

Meat also has significant biological and biochemical properties; meat iron presents a higher bioavailability than non-heme iron contained in plants or dairy products, and its absorption is largely unaffected by other dietary components; it also improves the absorption of non-heme iron [156].

Ruminant meat represents an important source of vitamin B12, represented in biologically active forms. Furthermore, the level of vitamin B12 stored in the liver and muscles of ruminants and synthesized before being absorbed is higher than in pork and chicken.

Finally, technological processes involved in meat preparation can significantly modify and influence both heme iron and vitamin B12 concentrations [157,158].

In the elderly, an “anabolic resistance” can be established towards the effects of hormones and nutrients, which results in a higher stimulation of muscle protein anabolism.

It is possible to set some nutritional strategies to improve and optimize the postprandial kinetics of amino acid absorption without changing the overall protein intake by concentrating the daily protein administration into one meal or taking on rapidly digested proteins [130,159,160].

A very important factor for the absorption and, therefore, the use of meat proteins is the chewing efficiency of patients; in fact, edentulous subjects, or those with poorly functioning prostheses or with chewing problems due to osteoarticular or neurological pathologies, swallow pieces of meat that are not very disintegrated, with a relative speed of digestion of proteins significantly reduced and, therefore, not easily digestible by the intestine, with a consequent reduced use of amino acids for the synthesis of body proteins [161].

In this case, a solution could be represented by the technological processing of the meat (mincing and cooking conditions), able to influence the digestion rate of meat proteins and their assimilation, thus, to counteract the problems of chewing inefficiency [162].

Two important amino acids are carnosine (β-alanine-L-histidine) and anserine (β-alanine-L-1-methyl-histidine), found exclusively in mammalian skeletal muscle, and are histidine-containing dipeptides (HDPs). Their concentration in meat is generally minimally influenced by technological processes [163].

Diets rich in HDPs can be beneficial for the elderly due to their antioxidant properties towards proteins and nucleic acids, due to their ability to bind divalent metal ions and trap free radicals; they also have an important buffer capacity. Furthermore, HDPs are able to reduce the aldehydes that are formed from unsaturated fatty acids during oxidative stress [164].

Furthermore, HDPs can safeguard against the glycation of proteins and cross-linking and, therefore, have an important role in protecting against neurodegenerative disorders such as Alzheimer’s disease and in the prevention of diabetic complications [165]; in fact, preventing the formation of the cross-link of proteins that interferes with their tissue function, HDPs do not perform their aggregation of cell material in the form of plaques [166,167].

A recent study using HDPs at the corresponding dose of about two servings of 120 meat in the diet of some elderly people showed a promising result on cognitive functioning [168].

Fish

The beneficial health effects of fish consumption have been attributed to the anti-inflammatory properties of n-3 long-chain polyunsaturated fatty acids (LC-PUFA), particularly eicosapentaenoic acid (EPA) and docosahexaenoic acid (DHA), which are present in high quantities especially in oily fish [169].

In fact, from the literature, it seems that the consumption of fish in the elderly is related to a better cognitive performance, to the improvement of memory in the case of mild cognitive impairment, to a reduction in symptoms of depression, to a lower risk of hip fractures, and to a better renal function [170,171].

A synergistic effect between selenium and n-3 LC-PUFAs on health has also been reported [172].

Unfortunately, old people consume insufficient quantities of fish and, even in the case of malnutrition, they tend to consume fatty fish of a poor nutritional quality.

Cereal-based foods

Carbohydrates have a special meaning in cereals, and usually contain around 50–80% carbohydrates on a dry-weight basis.

Starch is the most abundant cereal polysaccharide in diets and is an important source of energy. There are available carbohydrates, such as starch and soluble sugars, which are digested and absorbed by humans, and unavailable carbohydrates, i.e., dietary fibers, such as resistant starch, cellulose, and other complex polysaccharides (β-glucan, pectins, and arabinoxylans), which are not digested by the human intestine. These substances can be fermented by the intestinal microbiota, with the production of methane, carbon dioxide, water, and volatile fatty acids, or short-chain fatty acids (SCFA), essentially, acetic, propionic, and butyric acid, equipped with important functions: butyric acid supplies energy to the colocytes and integrates the mucosa, propionic acid is used for gluconeogenesis by hepatocytes, and acetic acid is used by different tissues as a substrate for the cycle of Krebs or intermediate in lipid synthesis [82,83].

In 2010, the EFSA determined that, based on the role of dietary fiber in bowel function, the adequate intake of daily dietary fiber is 25–35 g per day (of which 6 g is of soluble fiber) for regular intestinal function in adults [125].

In addition, the EFSA has approved some health claims regarding the consumption of dietary fiber:Arabinoxylan promotes the reduction in postprandial glycaemia [173,174];Barley grain, oat grain fiber, and sugar beet fiber promote the increase in fecal mass;Wheat bran fiber promotes intestinal transit and increases fecal mass [158,159,175,176].

With the wording “High fiber”, as listed in the annex to Regulation (EC) no. 1924/2006, it is understood that the product contains at least 6 g of fibers per 100 g or at least 3 g of fibers per 100 kcal.

During the last decades, epidemiological and clinical studies demonstrated that the consumption of dietary fiber and whole grain has a positive impact against obesity, type 2 diabetes, cancer, and CVD in middle-aged people [177,178,179].

The mechanism of action of dietary fiber and bioactive components that lead to these beneficial effects may be due to short-chain fatty acids such as propionate, coming from dietary fiber in the intestine, able to exercise, protective effects, a reduction in serum cholesterol linked to an increased rate of bile excretion, and a reduction in inflammatory mediators, such as interleukin-18 (IL-18) and C-reactive protein (CRP) [180,181].

In addition, adding additional ingredients to improve processing or to produce special breads can increase the nutritional value.

Bread is a fundamental source of essential nutrients such as carbohydrates, B vitamins (B1, B2, B3, B6, B9), and minerals (phosphorus, potassium, and magnesium) and, therefore, has a high nutritional value, so nutritionists indicate it as an essential constituent of the food pyramid.

In the United States, the Food and Drug Administration (FDA) has approved two health claims for dietary fiber. The first states that a diet that involves an increase in the consumption of dietary fiber, provided by fruits, vegetables, and whole grains, associated with a reduced fat consumption (<30% of calories), can reduce some types of cancer.

The “increase in consumption” is defined as ≥1 “equivalent ounce” (1 ounce = 28.3 g) with three ounces derived from whole grains [182].

The second FDA claim states that diets low in saturated fat (<10% of calories) and cholesterol, but high in fruits, vegetables, and whole grains, reduce the risk of coronary heart disease [183].

In summary, a healthy diet should consist of 25–35 g of dietary fiber per day, of which 6 g should be soluble fiber [184].

Fruits and vegetables

The anti-inflammatory properties of phytochemicals, in particular polyphenols, play an important role in their beneficial effects on healthy aging [185].

Several polyphenols have shown positive effects on cognition, including flavonoid-rich cocoa drinks or resveratrol [186,187].

In addition, polyphenols also have beneficial effects against other diseases associated with malnutrition, such as the catechins of green tea extract for sarcopenia and chocolate against chronic fatigue syndrome [188].

In patients with dysphasia, the ingestion of red wine (without alcohol) appeared to facilitate the swallowing reflex [189].

A recent review suggests that the use of flavonoids in post-menopause women promote the function of osteoblasts and reduce chronic low-grade inflammation; thus, protecting against bone loss and preventing osteoporosis [190].

An important key issue for the use of these bioactive phytochemical compounds useful for fortifying foods for the elderly is the evaluation of their bioaccessibility and bioavailability, especially with regard to the impact of food processing [191].

Water

Various factors contribute to the dehydration of the elderly; many of them do not reach their recommended daily intake of oral fluids, especially in the presence of diseases such as malnutrition and anorexia, or because of a decrease in appetite and less motivation to eat or drink, or a higher thirst threshold, typical of advanced age, or due to adverse effects of drugs, such as diuretics.

Older people are more vulnerable to water imbalance and, in the presence of dehydration, death occurs far more quickly than in the absence of any other nutrient, usually within a few days [192,193].

In old people, the consequence of suboptimal hydration has been associated with a greater susceptibility to urinary and pulmonary tract infections, pressure ulcers, confusion, and disorientation, while an adequate hydration results in less constipation, a better neuromotor performance with fewer falls, improved rehabilitation outcomes in orthopedic patients, and a reduced risk of bladder cancer in men [194].

Therefore, the best preventive measure to reduce the risk of dehydration in older people should be to consume at least 1.7 L of fluids over a 24 h period, to be increased by 500 mL per degree above 38 °C in the presence of a fever [195].

If dysphagia is present, it is necessary to administer fluids trough thickened beverages.

An important factor is also the availability of beverages and a greater choice of them [196], such as the availability of mineral drinks, milk, fruit and vegetable juices, and also sports drinks [195].

## 8. Conclusions

Malnutrition in the elderly has a complex and multifactorial origin; age is associated with physiological changes that cause the old person to be more vulnerable and more easily subjected to malnutrition in conjunction with risk factors, such as diseases, lifestyle, drugs, etc.

Gastrointestinal changes in aging involve motor function and therefore intestinal transit, mechanical breakdown of food and his chemical digestion. These alterations can progressively lead to the reduced ability to supply the body with adequate levels of nutrients, with the consequent development of malnutrition.

Recognizing risk factors is a fundamental step in being able to treat malnutrition.

An adequate protein intake associated with a correct energy intake is essential to prevent malnutrition and sarcopenia in the elderly; furthermore, if the diet is not sufficient, it is necessary to supplement any micronutrient deficiencies.

Literature studies show how the quality of the diet is very important with respect to the risk of frailty, and, in particular, the meta-analyzes have established that the Mediterranean dietary model has a high quality and greater beneficial effects, with a high intake of micronutrients and antioxidants, as polyphenols and bioactive compounds present in the diet, characterized by a high consumption of fruit, vegetables, and plant-based foods, use of olive oil, and lesser consumption of meat and dairy products [197,198,199,200].

However, there are still significant gaps in the literature regarding the evidence for the non-drug treatment of malnutrition, and it is necessary to plan high-quality prospective studies and well-designed clinical trials.

## Figures and Tables

**Table 1 nutrients-13-04337-t001:** Causes of malnutrition.

Physiological Causes
• Gastrointestinal diseases
• Dysphagia
• Malabsorption
• Oral problems (poor oral hygiene)/loss of smell or taste
• Respiratory diseases
• Endocrine diseases (diabetes mellitus)
• Neurological/psychiatric diseases
• Loss of autonomy with physical disability to feed self
• Infections
• Drugs interactions
• Cancer
• Poor appetite and poor diet
Psychological Factors
• Depression/anxiety
• Dementia/Confusion
• Alcoholism
Social Factors
• Loneliness and isolation
• Inability to prepare food and/or to shop
• Poverty

Modified by Rémond, D. et al., *Oncotarget* 2015, 6, 13858–13898 [38].

**Table 2 nutrients-13-04337-t002:** Changes with aging in gastrointestinal organ function and resulting alterations of nutrient status.

Organ Function	Change with Aging	Nutrient Alteration
Gastric acid secretion	Decreased with atrophic gastritis	Decreased absorption of folate and protein-bound vitamin B 12
Gastric motility	Slow liquid and mixed solid–liquid emptying. Preserved solid emptying	Decreased bioavailability of mineral, vitamins and protein
Small intestine structure and motility	Minor changes in structure.	No clinical significance
Small intestine microflora	Bacterial overgrowth in small bowel owing to atrophic gastritis	Increased bacterial synthesis of folate. Possible decrease fat-soluble vitamin absorption
Pancreatic secretion	Reduced capacity for bicarbonate and enzyme production	No clinical significance

Modified by Saltzman, J.R. and Russel, R.M. The aging gut, *Gastroenterol. Clin. N. Am*. 1998, 27, 309–324 [21].

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
