# Peer review of "Nutrition and Healthy Aging: Prevention and Treatment of Gastrointestinal Diseases"

_nutrients, 2021, doi:10.3390/nu13124337_

Round 1
Reviewer 1 Report
The narrative review provides a comprehensive overview on nutrition and aging, covering malnutrition, changes in the GI tract, and dietary strategies. It provides a wealth of information and can thus only remain on the surface in many sections. While this is unavoidable given the breadth of content, contradictions and inconsistencies in the text are troublesome. Especially in the last third of the manuscript, the text seems like a good first draft rather than a finalized text. The manuscript should be revised and finalized in its entirety and the inconsistencies and repetitions should be eliminated and the literature checked for correctness and up-to-datedness.
There are some major points (there are neither line nor page numbers in the MS, therefore, pages numbers refer to the page of the pdf document):
- Title: Seems misleading – the MS does not focus on prevention and treatment of GI diseases, it rather discusses the prevalence, causes and effects of malnutrition, physiologic changes in the GI tract and dietary strategies
- Page 2: The WHO (global nutrition) to my knowledge is the only international institution which takes “malnutrition” (bad nutrition) literally, and taxonomically defines adiposity as form of malnutrition. Clinical nutrition societies in Europe (ESPEN), the North-America (ASPEN), in Asia (PENSA), and Latin-America (FELANPE) equate malnutrition with undernutrition and discuss adiposity separately. As the manuscript is written by clinicians, it should take the position of the clinical societies, which is less confusing. The double burden of malnutrition and obesity in the elderly is still true and worthwhile to mention – although with corrected terminology
- The landmark paper of the first global malnutrition diagnostic criteria – the GLIM criteria – are not mentioned and should be added (Cederholm T et al, 2019, doi: 10.1016/j.clnu.2018.08.002)
- Citation 30 contains an error; correct is Clin Nutr 2017; 36(1):49-64 (see DOI: 1016/j.clnu.2016.09.004). The current citation mixes “Diagnostic criteria of malnutrition” published in 2015 (doi: 10.1016/j.clnu.2015.03.001) with the authors and title of the “Terminology in Clinical Nutrition of 2017
- Page 3: The current definition of sarcopenia is EWGSOP II, which is only mentioned later as cit 92 on page 9. Therefore, sarcopenia is NOT defined alone as loss of muscle mass, but by loss of muscle strength (first in the algorithm) and muscle mass (only second in the algorithm). It is recommended that the EWGSOP II criteria are mentioned early on page 3 to provide consistency
- Page 5: I think it should read, that cobalamin (vitamin B12) absorption is reduced and not riboflavin absorption (vitamin B2) – please check. There is also inconsistency with the related figure/table (here folate and vitamin b12). Folate is sometimes introduced as Vitamin B9 and sometimes as folate throughout the ltext – better to provide consistent wording
- Page 9 – para 5: citation is missing
- Page 10 – skin: It was long believed that hypoalbuminemia is caused by (and thereby, a feature) of malnutrition – but serum albumin – as a negative acute phase protein – is more affected by inflammation and does no longer count as malnutrition parameter (see Evans DC et al. Nutr Clin Pract. 2021;36:22–28). Therefore, delete hypoalbuminemia.
- Page 11: ONS is short for oral nutrition supplements not oral food supplements. You might also add “sip feeds” in parenthesis, as this term is better known
- Page 12 and the following: from here on the text seems like a first collection of thoughts and literature and should be more thoroughly worked out.
- Page 11: “need for nutrition assessment” sections describes only malnutrition screening tools but no assessment - or diagnosis tool – Therefore, reconsider the sub-heading
- Page 12: “Meat is a fundamental component of a balanced diet”… not true any more – vegetarian and vegan diets are considered as fully acceptable healthy diet by the major nutrition societies worldwide. Also plant proteins include essential (=indispensable) amino acids). Please provide a less opinionated and more balanced view on meat.
- Conclusion is missing and should be added
Author Response
Premise: the intent of the manuscript was to give practical messages that would be used in clinica practice.
The manuscript evaluates the role that the physiological changes tha occur with aging in the various gastrointestinal organs and that can affect malnutrition, however giving indications on treatment, mainly dietary; specifically treating the various patholohies would have entailed a further lengthening of the pathologies would have entailed a further lengthening of the revision.
The various comments proposed by the reviewer were reviewed and required changes made.
The review has been revised and corrected where repetitions were found, and literature controlled.
Conlcusions has been added.
Reviewer 2 Report
The aim of the review is to understand the pathophysiology of malnutrition in elderly in order to promote a healthy ageing but the manuscript sometimes is not well organized or clear in the different sections and there is lack of references. In addition, there is no discussion or conclusions and main changes should be performed before the article is suitable to be published.
The main concern is section 7 that should be completely rewritten: This section is not clear, the authors enumerate data from literature but do not discuss future approaches from a nutritional point of view to promote a healthy ageing sand prevent treat malnutrition. It would be highly desirable this section to rewrite and to divide in subheadings to better follow the manuscript as is has been performed in previous sections.
-It is important to mention the dietary guidelines from ESPEN, also the different ways to assess malnutrition, which are the standard ones, what does it consist each one on and explain why there is no consensus between studies. I don´t understand why the authors when they explain MNA use separate sentences as they were different measures. Please try to connect it better.
-The importance to the adherence to the Mediterranean diet in the elderly is poorly developed and recent data support the importance of this type of diet that should be indicated and referenced.
-This section should clearly indicate the importance of the different types of foods in elderly and what is the scientific evidence. There is lack of references. As it happens before in this section, the authors use separate sentences without connection. For example, different sentences related with meat… Later, the authors talk about carnosine and anxerine and it is confusing. Please check and rewrite all the paragraphs related to different types of foods.
-In the cereal based foods please better explain the importance of undigested complex carbohydrates that are fermented by the microbiota in host health and elderly. There is lack of references also in this section, for example in the documented effects of arabinoxylans and barley…
-Please enlarge and rewrite the section of fruits and polyphenols and connect the sentences.
-Please check the abbreviation for ONS and only mention the first time.
-Moreover, the authors should critical discuss the current strategies used for prevent and treat gastrointestinal diseases and malnutrition in aged people or what should be done in the future.
Specific comments:
-Introduction: Some sentences in this section are shaded in grey. Please correct.
-Section 2: Please use the same definition from sarcopenia in this section and later in section 6.2
-Section 3: Pathophysiology of undernutrition: This section is too sketechy and underdeveloped. It would be better to enlarge and follow the style from another sections and not only enumerate the different phrases associated with undernutrition. In addition, only a reference related to sarcopenia is indicated, please cite another ones.
-Section 5.3: The first sentence can be deleted because is later used through the section. Moreover later in the manuscript there is a table named changes with aging in gastrointestinal organ functions and resulting alterations. Please add the letter c at the beginning of the title. The authors indicate bacterial overgrowth in the small bowel and it would be necessary to better explain this disease in this section. I supposed they refer to SIBO but is not enough clear. Moreover, the employ of microflora is not appropriate and the word microbiota should be used instead.
-Section 5.4. It is too short and underdeveloped. It could be joined with section 5.6 microbial digestion.
In section 5.6: Regarding the bacterial metabolites, the SCFAs. It is important to briefly explain how they are produced, the name of main SCFAs and their biological functions. It is important also to mention that several recent publications demonstrated the reduction on the levels certain clostridia members (including important producers of butyric and and other SCFA, such as Faecalibacterium prausnitzii or Roseburia) and a reduction on other potentially beneficial microorganisms (Bifidobacterium).
-Aging has been related not only with reductions on microbial levels, but also to increase genus Akkermansia. In this section the authors should also highlight that the microbiota of elderly individuals reaching advanced age (centenariasn), is different, more diverse and similar to adults, than that of elderly not living so long.
Please check the font size of some phrases in this section and for the word bifidobacteria do not use capital letter.
-In section 5.7 the authors indicate that aging is associated with immunosenescence that is a progressive reduction of the mucosal immune response of the intestine but it not correct because immunosenescence is the decline of the immune function or the immune deregulation not only at intestinal level.
Please the names of the bacterial strains should be written in italics like H.pylori.
-Section 6: Please check the last sentence because to say that malnutrition represents one of the main geriatric syndromes and that is very frequent is redundant
-Section 6.1: It would be useful to indicate which are the main antioxidants and please do not write that vitamin E is a food antioxidant. Vitamin E is a micronutrient with antioxidant properties that can be present in certain foods and indicate dietary sources rich in Vitamin E
Author Response
Premise: the intent of the manuscript was to give practical messages that would be useful in clinica practice.
Section 7 includes sub-chapters, the message for healthy aging and preventing malnutrition seems clear enough to us.
The various comments proposed by the reviewer were reviewed and required changes made.
The review has been revised and corrected where repetitions were found, and literature controlled.
Conclusions has been added.
Round 2
Reviewer 1 Report
The authors implemented the suggestions and I have no further comments